# Evaluating Intraocular Pressure Alterations during Large Muscle Group Isometric Exercises with Varying Head and Body Positions

**DOI:** 10.3390/ijerph21040476

**Published:** 2024-04-13

**Authors:** Nina Krobot Cutura, Maksimilijan Mrak, Dominik-Mate Cutura, Ivanka Petric Vickovic, Lana Ruzic

**Affiliations:** 1Varazdin General Hospital, Ivana Mestrovica 1, 42000 Varazdin, Croatia; dominik.mate.cutura@student.kif.unizg.hr; 2Faculty of Kinesiology, University of Zagreb, Horvacanski zavoj 15, 10000 Zagreb, Croatia; lana.ruzic.svegl@kif.unizg.hr; 3Karlovac General Hospital, A.Stampara 3, 47000 Karlovac, Croatia; 4Department of Ophthalmology, Sestre Milosrdnice University Hospital Centre, Vinogradska cesta 29, 10000 Zagreb, Croatia

**Keywords:** isometric exercise, intraocular pressure, body position, ocular health

## Abstract

Performing physical exercise affects intraocular pressure, and its elevation and fluctuations are the main risk factors for glaucoma development or progression. The aim of this study was to examine the acute alterations in intraocular pressure (IOP) during four unweighted isometric exercises and to determine whether the different head and body positions taken during exercise additionally affect IOP. Twelve healthy volunteers between the ages of 25 and 33 performed four isometric exercises: wall sit in neutral head and body position, elbow plank in prone head and body position, reverse plank in supine head and body position for 1 min, and right-side plank in lateral head and body position for 30 s. Intraocular pressure was measured by applanation portable tonometry, before performing the exercise, immediately after exercise completion, and after five minutes of rest. A significant acute increase in intraocular pressure was found as a response to the performance of the elbow plank (*p* < 0.01), the reverse plank (*p* < 0.001), and the right-side plank (*p* < 0.001). The wall sit exercise did not reveal a statistically significant IOP elevation (*p* = 0.232). Different head and body positions had no significant additional influence on IOP (F (3,33) = 0.611; *p* = 0.613), even though the alteration in IOP was found to be greater in exercises with a lower head and body position. Our data revealed that IOP elevation seems to be affected by the performance of the elbow plank, the reverse plank, and the right-side plank; and not by the wall sit exercise. More different isometric exercises should be examined to find ones that are safe to perform for glaucoma patients.

## 1. Introduction

Engaging in regular physical exercise has significant health benefits, including the management and prevention of chronic diseases. Recently, isometric exercises have become more prevalent as they increase muscle strength and endurance without additional joint stress [1]. Isometric contraction occurs while exerting force against a fixed object or against an external load and does not involve any muscle or joint movement. Isometric exercises enhance the core stiffness, and it is known that core training is a crucial element of physical fitness regimens related to health [2,3]. Training programmes for athletes, but also army fitness tests, include numerous isometric exercises, such as planks [4]. As they are short and easy to perform, they are often part of recreational exercise programmes and home workouts performed by the general population, too. Given the benefits, they are also commonly used in injury rehabilitation programmes for patients [5].

However, research on effects of isometric exercises on different body systems has shown that they are not suitable for all individuals, particularly in terms of ocular health. The effect of isometric exercises on systemic circulation parameters is well known [6], but the effect on intraocular pressure (IOP) is often overlooked. Elevated intraocular pressure and its diurnal fluctuations are significant risk factors for the onset and progression of glaucoma [7]. Nowadays, glaucoma is one of the most common causes of irreversible vision loss in the adult population [8]. Intraocular pressure is the only modifiable risk factor for glaucoma; therefore, its regulation is crucial in the treatment of disease [9]. Numerous studies have examined the impact of different habitual and occasional activities, such as physical exercise, sleep-related postures, playing wind instruments, eye rubbing, and many others, on intraocular pressure changes. The findings demonstrated that IOP elevation and fluctuation episodes may occur during various daily activities, which is highly undesirable in glaucoma patients [10]. While an occasional short-term IOP fluctuation is likely to be insignificant, the cumulative effect of frequent fluctuations may have a substantial impact on glaucoma onset or progression [11].

Research on aerobic activities and strength exercises indicated that they are beneficial for glaucoma patients, being performed at a moderate intensity [12]. On the other side, studies of isometric exercise effects on intraocular pressure reported contradictory results. Risner et al. reported a general agreement that IOP decreases immediately after isometric exercises; thus, they were recommended for glaucoma patients [13]. However, some recent studies suggest that intraocular pressure remains unchanged or even increases with certain isometric exercises, especially when lifting weights or exercising at maximal exertion [14,15]. A study of intraocular pressure response to performing isometric exercises in high-load conditions reported a progressive IOP rise, regardless of subjects’ sex and type of exercise [16]. Also, it has been demonstrated that IOP increases continuously in parallel with systemic blood pressure during isometric exercise [17].

Different head and body positions taken in some daily activities, such as sleep-related postures or inverted body positions, have been shown to have a great impact on intraocular pressure, too. IOP measurements were higher in the supine position and right and left lateral decubitus positions. Sitting with the neck in the neutral position showed the lowest intraocular pressure [18]. A review comparing posture-induced IOP changes found that more pronounced IOP elevation occurs in inverted body and head positions, and that it seems to be dependent on the angle of tilt [19]. Considering that IOP peaks are detrimental in individuals with a high risk of glaucoma onset or progression, it is regarded that such head and body positions should be avoided in those groups. Regarding the different head and body positions taken during various isometric exercises, it is important to determine their association with intraocular pressure change as well.

Previous studies mostly examined an intraocular pressure response to weighted isometric workouts. As far as we are aware, the effect of isometric exercises using only body strength on IOP values has not been examined yet.

The aim of the present study was to examine the acute response of intraocular pressure to four unweighted isometric exercises, which are often performed as part of various training and rehabilitation programmes but also as home workouts by the general population. The secondary aim was to determine whether the different head and body positions taken during exercise additionally affected intraocular pressure. It was hypothesised that an immediate response of and increase in IOP will occur and that the lower head position during exercise contributes to the change to a greater extent. 

A health-promoting lifestyle that includes regular physical activity is highly recommended for people living with chronic conditions, and glaucoma patients should not be an exception. The findings from this research are aimed to help in the improvement of quality of life and wellbeing in the glaucoma population. Since reaching the target intraocular pressure and maintaining its stable values currently remains the only modality of glaucoma treatment [7,9], it is important to determine the exact effect of different daily activities, such as isometric exercises, on intraocular pressure.

## 2. Materials and Methods

### 2.1. Participants

The G*Power programme was used to calculate the required sample size. Using the published effect sizes from Vera, Redondo et al.’s research [16], the required number of subjects for a power of 0.80 was calculated for one group in five repeated measurements (F tests, repeated measures ANOVA, “within analysis”). In the mentioned research, the effect size for several different activities ranged from −0.53 to 1.62. Based on that, in the a priori determination of the required number of subjects, the average effect size of 0.505 was included. As it was planned to measure IOP values at rest and then repeat measurements in two different situations, the minimum required number of subjects was calculated to be eight. Due to the possibility that in this research the effect size would not be so large, the required number of subjects was increased by 50% (twelve subjects).

Seven women and five men between the ages of 25 and 33 participated in the study (28.67 ± 2.39). All subjects were free of systemic and ocular disease. Ophthalmological exclusion criteria were baseline intraocular pressure >21 mmHg, contact lens wear, refractive surgery procedure, and a refractive error >±3.00 diopter sphere (DS) and ±1.50 diopter cylinder (DC). The level of physical activity had to be less than 3 days/week of light or moderate intensity activity in order to exclude high-fitness individuals. According to Vera et al., fitness level modulates the IOP response to strength exercise, with high-fitness individuals showing a more stable IOP response [20]. The fitness level in this study was determined by an interview about the implementation of physical exercise in the past 12 months. Exercise frequency and self-assessment of exercise intensity were determined. Subjects with locomotor limitations were excluded. Subjects were instructed not to consume caffeine, alcohol, or cigarettes on the day of the measurement. Women did not participate during the menstruation phase. All subjects were informed about the purpose, benefit, and risk of this study and gave informed consent for inclusion before they participated in the study. The study was conducted in accordance with the Declaration of Helsinki, and the protocol was approved by the Ethics Committee of the Faculty of Kinesiology, University of Zagreb (96/2023).

### 2.2. Study Design

Firstly, all subjects underwent a complete ophthalmological examination. If they met the inclusion criteria, their physical activity level was determined by a short interview. They were then instructed on how to perform four different isometric exercises. The ability to perform exercises within the given time was tested in the first session. If no limitations were found, subjects were scheduled for the research measurements the next day. The second session consisted of the performance of exercises and intraocular pressure measurements on the right eye at three time points for each exercise, as described in Section 2.2.2 and Section 2.2.3.

#### 2.2.1. Ophthalmological Examination

Initial examination included determination of best corrected visual acuity, slit lamp biomicroscopy of the anterior segment of the eye, baseline IOP measurement by applanation tonometry, and fundoscopy of the optic nerve head and macula. Central corneal thickness (CCT) was measured by anterior segment optical coherence tomography (AS-OCT) to ensure that it did not alter the IOP value.

#### 2.2.2. Isometric Exercises

Four different unweighted isometric exercises with different head and body positions were performed: wall sit in neutral head and body position, elbow plank in prone head and body position, reverse plank in supine head and body position, and right-side plank in lateral head and body position. Considering that the IOP measurements in all exercises were planned to be carried out on the right eye only, the right-side plank was chosen over the left-side plank. According to Malihi and Sit, the IOP value is higher in the dependent eye when measured in the lateral decubitus positions (meaning the right eye in the right decubitus position) [18]. Thus, the right-side plank was chosen so that the right eye would be a dependent one. The wall sit, elbow plank, and reverse plank were performed for 1 min, while the right-side plank was performed for 30 s. Subjects followed a normal breathing pattern, and holding their breath was not permitted due to the possible effect of Valsalva manoeuvre on IOP. Exercises were performed in a randomised order, and there was a five-minute rest period between each for muscular recovery. Intraocular pressure was measured at three different time points according to the protocol described in Section 2.2.3.

#### 2.2.3. Intraocular Pressure Measurement

A Perkins Mk3 applanation tonometer (Haag-Streit, Bishop’s Stortford, UK) was used to assess intraocular pressure, as it is a hand-held device that can be used in different body positions. It is considered the “gold standard” for portable tonometers. The measurement was carried out by probe contact with the centre of the cornea while the subject fixated a distant object. One minute prior to the measurement, a topical anaesthetic drop (tetracaine) and a fluorescein dye drop were instilled into the examined eye. All measurements were carried out by the same investigator, on the right eye, and in accordance with the manufacturer’s recommendation. First, intraocular pressure was measured before performing the exercise (IOP 1) and then remeasured immediately after exercise completion (IOP 2). The third IOP measurement was made after 5 min of rest (IOP 3). The mean values of three consecutive IOP measurements were considered. Measurements were assessed in the afternoon (between 1 p.m. and 2 p.m.) to avoid the influence of circadian rhythm on IOP. 

### 2.3. Data Analysis

The software package STATISTICA 14.0 for Windows was used to process the data. The normality of the data distribution was tested using the Shapiro–Wilk test. Descriptive statistics were presented in the form of arithmetic mean, standard deviation, minimum, and maximum. Repeated measures analysis of variance (ANOVA) was used to analyse differences in IOP values at three different time points. ANOVA was used after inspecting variability plots for their robustness against slight normality violations. A repeated measures ANOVA was also used to determine the differences between IOP values in four different head positions immediately after contraction. A Fisher LSD post hoc test was used for further analysis of statistically significant data. The level of statistical significance was set at *p* < 0.05.

## 3. Results

The anthropometric characteristics of the participants are described in Table 1. 

The mean refractive error of the participants was −0.58 (±0.89) DS and −0.15 (±0.38) DC. The mean central corneal thickness was 542.67 (±21.66) µm.

The mean values and standard deviation of IOP at three time points in four different isometric exercises are described in Table 2.

A statistically significant increase in intraocular pressure (IOP 2) compared with the IOP value before exercise (IOP 1) was observed immediately after the completion of the elbow plank (*p* < 0.01), reverse plank (*p* < 0.001), and right-side plank (*p* < 0.001). After a 5-min rest, there was a statistically significant IOP lowering (IOP 3) in comparison with IOP 2 in the elbow plank (*p* < 0.001), reverse plank (*p* < 0.05), and right-side plank (*p* < 0.01). IOP 3 did not differ significantly from the IOP value before exercise (IOP 1) in any exercise (elbow plank *p* = 0.146; reverse plank *p* = 0.157; right-side plank *p* = 0.212). 

### 3.1. Elbow Plank

Figure 1 demonstrates a significant increase in IOP immediately after the elbow plank exercise. A post hoc LSD test revealed that IOP 2 values differed significantly both from baseline (IOP 1) and from five-minute rest values (IOP 3).

### 3.2. Reverse Plank

Figure 2 shows a statistically significant intraocular pressure elevation (IOP 2) upon completion of the reverse plank exercise. In post hoc analysis, IOP 2 differed significantly from baseline IOP (IOP 1) as well as from the IOP values observed after 5 min of rest (IOP 3).

### 3.3. Right-Side Plank

In Figure 3, the significant IOP increase is visible immediately after the right-side plank exercise. As in the elbow plank and reverse plank exercises, the post hoc LSD test observed a statistically significant IOP 2 alteration both from IOP 1 and IOP 3.

### 3.4. Wall Sit

At the end, the fourth exercise in an upright position of the head, the wall sit exercise, did not elicit a significant increase in intraocular pressure like the previous three exercises (*p* = 0.232), Figure 4.

### 3.5. The Head Position

When we analysed the influence of head position during exercise, we compared the immediate response in IOP in all four exercises. The whole model was not statistically significant, even though the wall sit was the only exercise that did not elicit a significant increase in IOP from rest (F (3,33) = 0.611; *p* = 0.613). Figure 5 demonstrates that IOP values measured after the wall sit were not as high as the IOP values in the other three exercises, with head positions much lower.

Post hoc power analysis was performed for notable findings, meaning exercise, in which a significant increase between rest and exercise was confirmed. The effect sizes were calculated in the free G*Power programme and then used to calculate the obtained power given a beta to alpha ratio of 4 (alpha 0.05 and beta 0.20). The obtained power was satisfactory and varied from 0.79 for the elbow plank and 0.94 for the reverse plank to 0.97 for the right-side plank, which seemed to elicit the strongest response and effect size in IOP. The obtained effect sizes of 0.75 to 1.5 suggest that the observed effect may be of practical significance.

## 4. Discussion

The purpose of this study was to assess the acute impact of different unweighted isometric exercises in different head and body positions on intraocular pressure values. The main finding is that unweighted isometric exercise indeed induces a significant increase in IOP. There was an immediate increase in IOP as a response to the performance of the elbow plank, reverse plank, and right-side plank. Conversely, there was no significant change in IOP as a response to wall sit. During a five-minute rest, IOP returned close to baseline values in all exercises.

The results obtained for the elbow plank, reverse plank, and right-side plank are consistent with some of the previous studies that found an acute IOP elevation when performing isometric exercise [16,17,21]. However, most of these studies included only high-load isometric exercises, making the results difficult to compare. It is well known that the performance of exercise against heavier loads results in a greater IOP elevation [16]. Comparatively, the greatest IOP elevation observed in this study was approximately 2.5 mmHg, whereas in a study on high-load isometric squat exercise of the same duration, the increase was approximately 8 mmHg after the workout was completed [22]. The reason for the slighter, but still significant, IOP increase in this study could be due to performing isometric exercise with no weight load. 

Even though increased IOP was present for a short period of time, it is known that short-term IOP fluctuation leads to a glaucoma progression, too [23]. Short-term IOP fluctuation refers to the difference between the highest and lowest IOP within a 24 h period [24]. Therefore, it seems that normal IOP diurnal fluctuation might be affected by performing these exercises. Taken together, such outcomes suggest that when a stable IOP is desired, these types of isometric exercises should be avoided, especially in high-load conditions.

On the other side, performing a wall sit did not significantly affect IOP. Certain studies in which subjects performed squatting for 6 min observed similar findings [25,26]. One of the possible reasons for such a result could be that the lower extremity muscles mostly participate in the performance of wall sit. According to some studies, the predominant involvement of the lower limb muscles in strength exercises does not result in a significant increase in IOP [27,28]. Also, maintaining the neutral head position while performing the exercise could be the reason why intraocular pressure was not significantly affected during the wall sit. The isometric exercises that involve predominantly upper extremities while remaining the neutral head position, such as the wall press or hanging hollow hold, could be useful in the differentiation of the potential cause.

This finding indicates that not all isometric exercises are contraindicated for glaucoma patients and individuals at risk for glaucoma, as previously suggested. Thus, it can be assumed that wall sit would be a relatively safe choice of exercise for individuals at risk of glaucoma onset and progression.

Regarding different head and body positions, exercises with the lower head position did not change IOP values to a great extent, but still, the change was significant in three of them, while in the wall sit exercise it was not. That might raise the question of choosing the right exercises for glaucoma patients in, for example, programmes like yoga, in which many different and inverted body positions are involved. As far as we are aware, only one study, which was published during the conduct of our research, compared IOP in different body positions while performing exercises. In that study, IOP alterations in supine and seated positions were compared during the performance of the bench press [29]. Comparable to our results, performing exercise in the supine position resulted in a greater IOP rise than in the seated position.

In our study, the novelty was the comparison of IOP changes in not only the supine and seated positions but also in the prone and lateral body positions. Until now, various studies of IOP changes in different body postures have carried out measurements only at rest, observing that mean IOP while sitting was significantly lower than in different recumbent positions, including supine, prone, and lateral decubitus positions [18,30]. Sultan et al. found an increase in episcleral venous pressure in a recumbent position, which is directly related to an IOP elevation [31]. Minor IOP differences in this study, in comparison with previous similar publications, may be caused by the shorter duration of a posture change during the exercises. 

The relevance of present study is manifested in the selection of interventions that are common in everyday practice but have not been examined in controlled experimental conditions so far. The elevation of IOP has been found to positively correlate with applied weight loads in isometric exercises [22], but this is the first study to demonstrate the IOP changes when using no weight load and in different body positions. Our findings incorporate novel perspectives regarding the impact of unweighted isometric workouts on IOP level as well as the significance of the head and body positions utilised during the exercises. 

Through further research concerning the impact of different exercise modalities on intraocular pressure, it could be possible to determine potential activity limitations for individuals at risk of glaucoma and glaucoma patients. The presented results show that some exercises should be avoided to prevent IOP fluctuations, which are a major risk factor in the disease’s onset and progression. Based on this, the guidelines for physical activity modulation could be determined, and an individualised approach could be applied in the prevention and treatment of glaucoma.

The main limitation of this study is that the experimental sample consisted of only healthy, young subjects, and these findings cannot necessarily be implicated in glaucoma patients or individuals at risk for glaucoma. Similar, or even greater, IOP changes are expected in those groups due to the greater susceptibility of the optic nerve head to mechanical damage. Also, glaucoma affects a predominantly elderly population, and an age-related alteration in the optic nerve head additionally increases the susceptibility to biomechanical forces exerted by elevated IOP [32]. Future studies should examine more different unweighted isometric exercises on a larger sample, including glaucoma patients, to find exercises that do not significantly affect desirable IOP levels. However, it should be considered that any increase in IOP during exercise, which would have no adverse effect on healthy individuals, may lead to new or additional glaucomatous change in susceptible individuals.

## 5. Conclusions

The performance of the unweighted isometric exercises in different head and body positions can cause an acute increase in intraocular pressure after the completion of the exercise. The elevated intraocular pressure was found immediately after the elbow plank in prone head and body position, the reverse plank in supine head and body position, and the right-side plank in lateral head and body position in healthy individuals. The wall sit exercise in neutral head and body position did not cause a significant change in intraocular pressure, so it could be considered a safer choice for exercise in individuals with glaucoma onset or progression risk. The intraocular pressure values returned to baseline levels within five minutes of rest in all four exercises. Our findings highlight that, even though heavy loads are not used during the exercise, caution should be taken when IOP fluctuations are undesirable. Isometric exercises with neutral head and body positions should be preferred over other possible positions. Health-promoting behaviour, such as regular physical activity, is an important part of chronic disease management, which also applies to glaucoma patients. It is crucial to determine whether the isometric exercises that are found not to raise IOP in a healthy population can be considered safe in a glaucoma population, too. The present outcomes indicate the need for individualised exercise programmes to reduce the risk of developing glaucoma and prevent disease progression in high-risk individuals.

## Figures and Tables

**Figure 1 ijerph-21-00476-f001:**
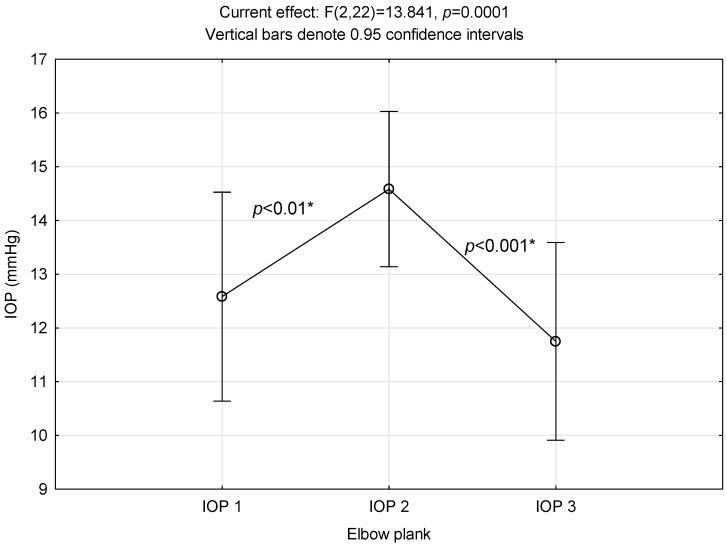
Intraocular pressure alterations related to elbow plank performance. *: significant difference.

**Figure 2 ijerph-21-00476-f002:**
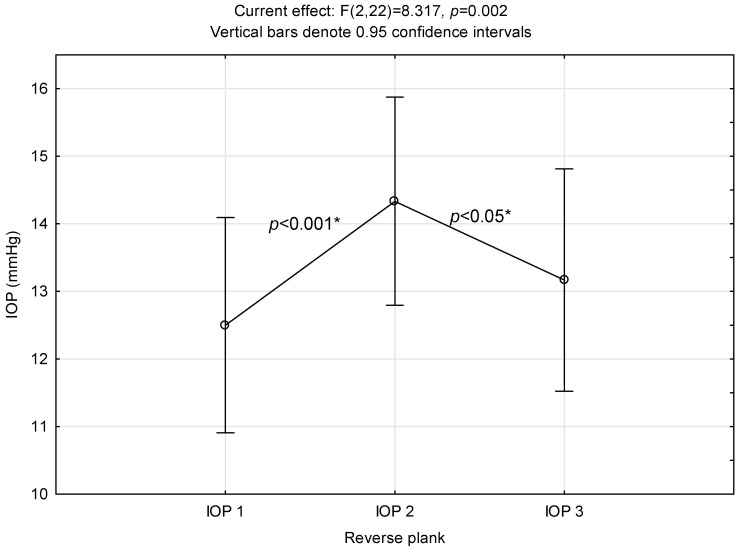
Intraocular pressure alterations related to reverse plank performance. *: significant difference.

**Figure 3 ijerph-21-00476-f003:**
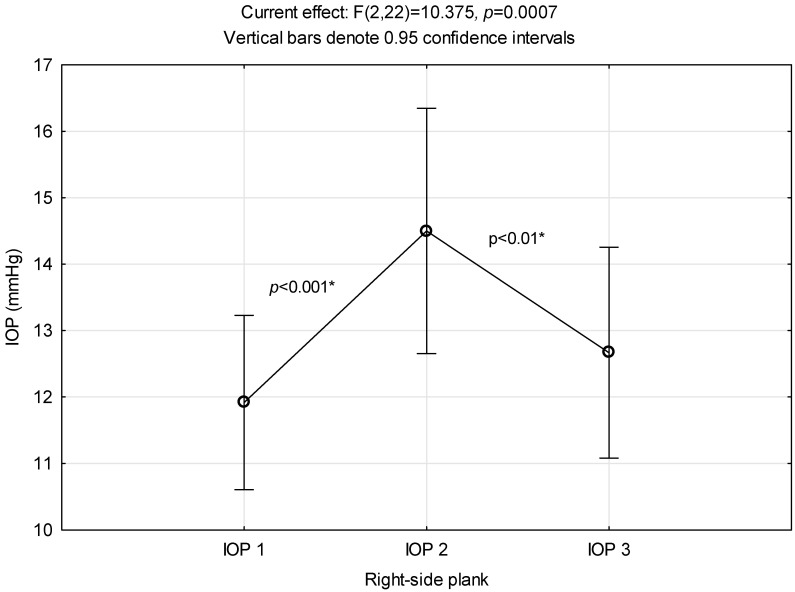
Intraocular pressure alterations related to right-side plank performance. *: significant difference.

**Figure 4 ijerph-21-00476-f004:**
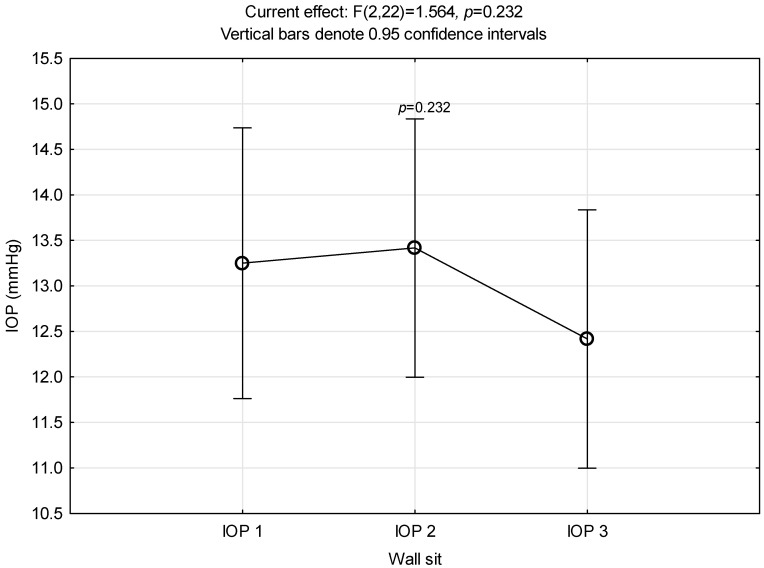
Intraocular pressure alterations related to wall sit performance.

**Figure 5 ijerph-21-00476-f005:**
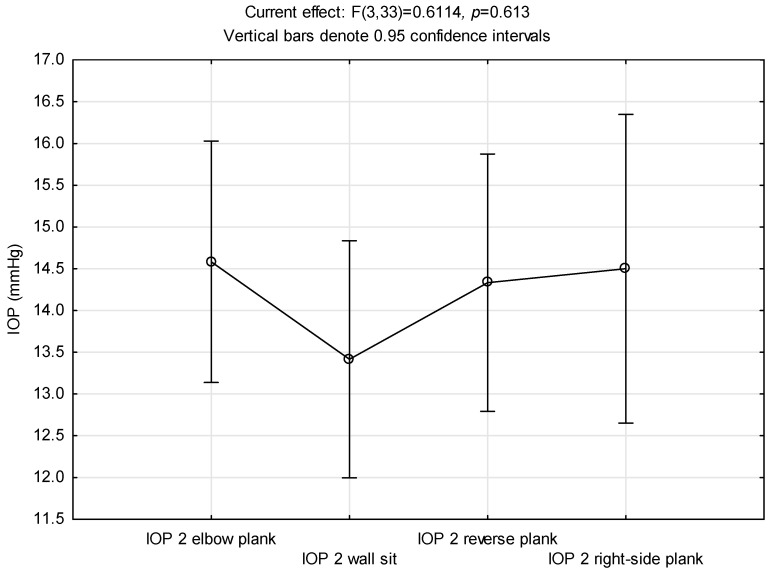
Comparison of intraocular pressure values immediately after the exercises’ completion.

**Table 1 ijerph-21-00476-t001:** Anthropometric characteristics of participants (mean ± standard deviation).

	Women (*n* = 7)	Men (*n* = 5)
Age (y)	29.00 ± 2.65	28.2 ± 2.17
Height (cm)	169.00 ± 7.48	182.6 ± 5.08
Weight (kg)	64.00 ± 7.16	84.2 ± 5.36
BMI (kg/m^2^)	22.43 ± 2.58	25.36 ± 2.68

**Table 2 ijerph-21-00476-t002:** Intraocular pressure (IOP) values in mmHg before exercise, after exercise, and after five minutes of rest (mean ± standard deviation).

	IOP1 *	IOP2 **	IOP3 ***
Wall sit	13.25 (±2.34)	13.42 (±2.23)	12.42 (±2.23)
Elbow plank	12.58 (±3.06)	14.58 (±2.27)	11.75 (±2.90)
Reverse plank	12.50 (±2.50)	14.33 (±2.42)	13.17 (±2.59)
Right side plank	11.92 (±2.06)	14.50(±2.91)	12.67 (±2.50)

* IOP1: value before exercise; ** IOP 2: value after exercise completion; *** IOP 3: value after five-minute rest.

## Data Availability

The data used in the present study are available from the corresponding author upon reasonable request.

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
