# Peer review of "Evaluating Intraocular Pressure Alterations during Large Muscle Group Isometric Exercises with Varying Head and Body Positions"

_ijerph, 2024, doi:10.3390/ijerph21040476_

Round 1

Reviewer 1 Report

Comments and Suggestions for Authors

Here authors have evaluated the change in intra ocular pressure with different isometric exercises which are risk factors in the progression of glaucoma. Accordingly, authors have found a significant increase in IOP in plank exercises except wall sit. Data suggests change in IOP is in relation to the position of head. Results presented in the study is of importance to subjects suffering from glaucoma due to increased intraocular pressure. The reviewer has following concerns.

Small sample size undermines the significance of the results presented. Also, study subjects belong to a healthy group of comparatively younger age. What is the effect of these positions in glaucoma patients with elevated IOP? IOP tend to increase with age, does it play in a similar fashion in older age individuals with high prevalence of glaucoma?

Table-2: There are differences in IOP1 (value before exercise) values taken before four different isometric exercises. Is there any statistically significant difference among those IOP1 values?

Authors can elaborate more on the clinical relevance of the presented results.

p-values are not presented in the figure-3 and 4 between IOP values as shown in figures-1 and 2.

Comments on the Quality of English Language

Minor editing of English language required.

Reviewer 2 Report

Comments and Suggestions for Authors

Comment: In this study, the authors measured intraocular pressure (IOP) during four unweighted isometric exercises to assess the impact of various head and body positions on IOP. The findings from this study may support future decisions regarding the selection of exercises deemed safe for individuals with glaucoma. Below are several points of consideration that require attention before publication.

Question 1: In the introduction section, it is stated that "...IOP increases continuously in parallel with systemic blood pressure during isometric exercise." Has systemic blood pressure been measured in parallel with IOP in this study? It is recommended to include this data if available and correlate it with the IOP results.

Question 2: In the materials and methods section, it is detailed that the level of physical activity of eligible participants had to be less than 3 days/week of moderate-intensity activity. What is the rationale behind this criterion? Why were participants who exercise more than 3 days/week excluded? Similarly, why wasn't a minimum level of exercise/week included as a requirement (minimum fitness level)? It seems that comparing non-exercising and active individuals may not be a fair comparison.

Question 3: It is suggested to include the reasoning behind choosing to perform a right-side plank over a left-side plank, or even both. Additionally, since IOP has been measured in the right eye, would the authors expect different IOP results if the measurement had been taken in the left eye instead, during a right-side plank? Please elaborate on this.

Question 4: Figure 3 seems to be missing p-values in the graph, as were added in previous graphs.

Question 5: In the discussion section, the authors mention that "It is known that an IOP elevation of 1 mmHg increases the risk of glaucomatous progression by up to 10%." However, the study cited by the authors (reference 22), upon which this statement is based, concluded that each mmHg of IOP on follow-up was associated with an approximate 10% increase in the risk of progression, but this follow-up occurred after 3 months. In this work, increased IOP is present over a very short period corresponding to the duration of the exercise, after which IOP returns to the baseline value. Therefore, it is recommended to add literature supporting the idea that slight and sporadic increases in IOP indeed lead to the progression of the disease.

Question 6: It is unclear whether the authors believe that IOP did not change significantly in the wall sit exercise due to only lower extremity muscles being involved or because it was a neutral head position-type of exercise. Would the inclusion of other exercises help in answering this question? If so, which exercises and why? Please elaborate on this in the manuscript.

Question 7: The authors pointed out that a main limitation of their study was that the experimental sample consisted only of healthy and young subjects. It is recommended to elaborate what the authors would expect to be the differences in IOP changes with exercising glaucoma patients, as compared to the current results of this study. Relevant literature should be added to support the authors’ opinion.

Round 2

Reviewer 2 Report

Comments and Suggestions for Authors

All the requested revisions have been adequately addressed.